# Tandem: a Confidence-based Approach for Precise Medical Image Segmentation

### Simone Monaco
simone.monaco@polito.it
Department of Control and Computer Engineering,
Politecnico di Torino
Torino, Italy

### Lorenzo Petrosino
lorenzo.petrosino@unicampus.it
Department of Engineering, Research Unit of Computer
Systems and Bioinformatics
Roma, Italy

### Christodoulos Xinaris
christodoulos.xinaris@marionegri.it
Istituto di Ricerche Farmacologiche Mario Negri - IRCCS
Bergamo, Italy

### Daniele Apiletti
daniele.apiletti@polito.it
Department of Control and Computer Engineering,
Politecnico di Torino
Torino, Italy

## ABSTRACT

The advent of Healthcare 4.0 has heralded a disruptive change in medical diagnostics, with Artificial Intelligence (AI) playing a central role in improving diagnostic accuracy and treatment efficacy. This work addresses the integration of AI into medical imaging, specifically through the use of deep learning networks to analyze and segment medical images with high precision. Our study builds on the capabilities of U-Net-like models to address the challenges of segmenting clinical targets of different sizes, such as cysts and tumor masses, in medical images. We present a novel deep learning segmentation solution, Tandem (**T**andem **A**nalysis for **N**eural **D**etection and **E**valuation **M**odel), which combines a segmentation model with a classifier to produce a confidence map alongside the model's prediction. This approach aims to improve segmentation accuracy by refining predictions and providing a mechanism to assess the reliability of the model, especially when identifying smaller clinically significant targets. We evaluate our method across various imaging modalities, including 2D and 3D acquisitions. We focus on detecting and segmenting kidney cysts associated with Autosomal Dominant Polycystic Kidney Disease (ADPKD) and tumor masses. The practical effectiveness of Tandem is demonstrated by the generation of reliable confidence maps that help clinicians make informed diagnostic and treatment decisions. This study represents a significant step towards precision medicine by improving the diagnostic capabilities of AI-driven systems in medical imaging.

## KEYWORDS

Medical image segmentation, small-objects detection, deep-learning models confidence

**ACM Reference Format:**
Simone Monaco, Lorenzo Petrosino, Christodoulos Xinaris, and Daniele Apiletti. 2024. Tandem: a Confidence-based Approach for Precise Medical Image Segmentation. In *ACM KDD 2024 Workshops, Artificial Intelligence and Data Science for Healthcare: Bridging Data-Centric AI and People-Centric Healthcare, August 25–29, 2024, Barcellona, Spain.* ACM, New York, NY, USA, 6 pages.

## 1 INTRODUCTION

In the healthcare 4.0 landscape, patient data has become an invaluable resource for clinicians seeking to make more accurate diagnoses. The integration of Artificial Intelligence (AI) is at the forefront of this revolution, significantly improving diagnostic processes and therapeutic decisions. Among the different types of data, neural networks have excelled in the field of medical image analysis [9, 16, 18].

The ability of AI-driven systems to analyze vast amounts of image data with high accuracy is transforming medical diagnostics. These systems can recognize nuances in medical images that are sometimes imperceptible to the human eye, enabling early and accurate disease detection, which is critical for effective treatment. The integration of AI into medical imaging goes beyond simple analysis; it extends to the development of advanced Decision Support Systems (DSS) [17]. These systems provide clinicians with invaluable insights and offer evidence-based recommendations that aid both diagnosis and the selection of optimal treatment strategies.

The advancement of AI technologies and machine learning algorithms and their integration into medical imaging and decision support systems is ushering in a new era of precision medicine, enabling more accurate diagnoses and treatments [11].

Furthermore, this technology has shown exceptional proficiency in detecting complex medical conditions, such as cysts in tissues affected by Autosomal Dominant Polycystic Kidney Disease (ADPKD), as well as tumor masses. There are still challenges because these objects come in a wide range of sizes, which makes it difficult for neural networks. In this situation, small objects have little impact on the traditional cost function used to train the models, which makes them likely to ignore clinically significant objects.

Based on previous studies that have used U-Net-like models to segment variable-sized clinical targets, we aim to enhance these

models with an automatic confidence estimation tool. Our approach involves a flexible architecture integrating prediction refinement into deep segmentation models. This aims to reduce prediction errors and assist clinicians by providing a mechanism to measure the model's confidence in its predictions, regardless of the size of the objects being analyzed and across different regions of the images.

In our study, we emphasize the limitation of the current solutions by focusing on the segmentation of cysts engineered using cells from ADPKD patients and the detection of tumor masses in liver tissues. For the first target, we utilized a dataset from the *Istituto di Ricerche Farmacologiche Mario Negri IRCCS*, containing RGB immunofluorescence images of human tubules engineered from cyst-lining cells. This research builds upon the work of Monaco et al. [7]. For the second target, we selected the well-known Liver Tumor Segmentation Benchmark (LiTS) [2], which consists of CT liver 3D images.

Summing up, this work aims to improve segmentation accuracy for clinical objects by adapting deep learning segmentation solutions to provide a size-invariant self-assessment of their predictions. The main contributions of this study are summarized as follows:

- Introduction of a novel segmentation solution, Tandem (**T**andem **A**nalysis for **N**eural **D**etection and **E**valuation **M**odel), which integrates a classifier with a segmentation model jointly trained to produce a confidence map alongside the usual model prediction.
- Evaluation of the proposed strategy on two different medical image segmentation tasks involving 2D and 3D acquisitions.
- Practical evaluation of the Tandem procedure to generate a reliable confidence map of the predictions to assist clinicians in model evaluation.

## 2 RELATED WORKS

The integration of AI in healthcare, particularly in medical image analysis, is the subject of extensive research. In the context of Healthcare 4.0, significant progress has been made in AI-driven diagnostic tools that improve the accuracy and efficiency of medical diagnoses. Several important studies have laid the foundation for these developments, particularly in the context of medical image segmentation and disease detection.

The use of AI in medical image analysis is well documented, with neural networks playing a central role. Early work by LeCun et al. [8] demonstrated the potential of convolutional neural networks (CNNs) in image classification tasks, which have since been adapted for medical imaging. Studies have shown that AI systems can accurately recognize and classify various medical conditions from image data. For example, Esteva et al. [3] successfully applied deep learning to dermatology and achieved the dermatologist-level classification of skin cancer.

In the field of kidney diseases, particularly ADPKD, deep learning has been used for cyst detection and segmentation. The U-Net architecture introduced by Ronneberger et al. [4] has become a cornerstone of medical image segmentation due to its ability to learn from relatively small datasets. As the number of U-Net-based architectures grew on various applications, different studies proposed comparisons of these new techniques to the kidney or cyst

segmentation task [5, 13, 20]. Most of the studies have concentrated on segmenting the entire organ to measure the overall presence of cysts. While this provides valuable information about the disease state, it does not offer a complete picture of the situation, which could be achieved by segmenting the cysts within the kidney tissue itself. However, this approach presents challenges due to the considerable variation in the size of these cysts. Recent advancements have focused on enhancing the accuracy of segmenting small and scattered medical targets. Monaco et al. [7] highlighted the limitations of many U-Net variants in segmenting smaller cysts in ADPKD and emphasized the necessity for new techniques to address or at least quantify this issue. This issue has also been addressed in a recent study [12], where the researchers investigated how this problem could impact the measurement of a treatment's effectiveness. They found that deep learning models showed similar behaviour across various treatments. However, the issue still persists.

In this context, Shirokikh et al. [14] investigated the performance of various loss functions for detecting objects of different sizes in the context of tumor detection on 3D CT images. To this end, they introduced a loss reweighting technique applicable to numerous loss function families to increase the penalty for errors on smaller objects. However, this approach remains relatively unexplored in the literature, where the problem is typically addressed through specific modifications to the model architecture [15, 21]. Other studies have tackled the issue in liver tumor segmentation by using contour-based loss functions to preserve segmentation boundaries [10], or by managing heterogeneous image resolutions with a multi-branch decoder [19].

Another critical aspect of AI in medical imaging is confidence estimation, which deals with the reliability of AI predictions. Kohl et al. [1] introduced a probabilistic U-Net, which provides segmentation results and estimates the uncertainty of these predictions. This approach is particularly useful in medical diagnostics, where understanding the confidence level of detection can significantly influence clinical decisions. However, the uncertainty generated by the Probabilistic UNet is not always easy for clinicians to interpret, making it less straightforward to apply in practical contexts to alert them to possible local model failures. Our method aims to provide more directly interpretable confidence scores, which can be more immediately useful in clinical decision-making.

## 3 MATERIALS AND METHODS

To improve medical segmentation models and ensure reliable self-evaluation of their predictions, we propose Tandem, an end-to-end pipeline for enriching the output of a neural network tailored to segmentation tasks. This is achieved by a *self-evaluation* head that provides a confidence measure for different parts of the prediction. Figure 1 illustrates the main components of our framework, which include a segmentation neural network and a classifier to estimate the error of the different parts of the original prediction. The key concepts of this architecture are explained in more detail in the following section.

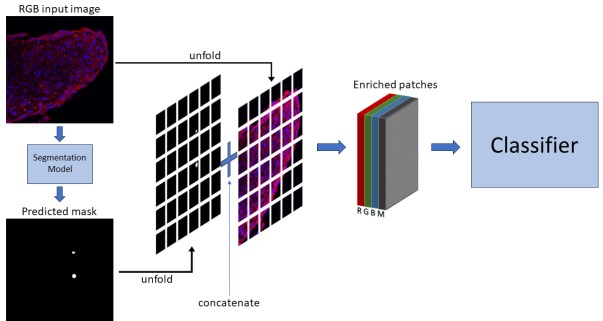

Figure 1: Pipeline diagram of Tandem

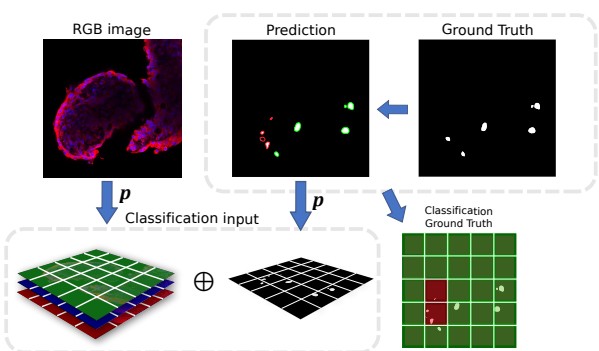

Figure 2: Visual representation of the generation process of classifier inputs and ground truths.

## 3.1 Tandem architecture

As previously mentioned, our approach uses classification to improve segmentation results. Our pipeline is not architecture-specific, so the two neural networks for segmentation and classification can be selected according to the specific requirements of the use case.

Starting with an input image of size $D \times H \times W \times C$ (where $D$ is included for 3D images, $H$ and $W$ are the other spatial dimensions, and $C$ is the number of channels), the first model $S(\cdot, \theta_s)$ produces a prediction with the same spatial dimensions and $C'$ channels, representing the number of classes to segment.

The classifier $C(\cdot, \theta_c)$ acts then as a sliding window, scanning the segmentation prediction along with the associated input image to detect inconsistencies. It refines the segmentation output by merging its predictions, thus contributing to the final segmentation result. To scan the input image with the classifier, we divide it into non-overlapping square patches of dimensions $d \times h \times w$ (where $d$ is included for 3D images) such that the dimensions along each axis are divided evenly into the corresponding original image dimensions.

The classifier's input is a tensor of size $d \times h \times w \times (C+1)$, where the channels consist of the original image channels concatenated with the first step's prediction. We refer to these 4-dimensional patches as enriched patches. These enriched patches are then passed to the classifier, which predicts a label for each patch, considering the segmentation prediction. The classifier's output is then a binary prediction for each patch of the input image, which is compared with a ground truth label according to the following rule:

- **True**: If either the prediction detects a target that is present in the patch ground truth, or if the model finds no target and there is indeed nothing in the ground truth.
- **False**: If the prediction detects a target that is not present in the ground truth, or if the model fails to detect a target.

A target is marked as *detected* if the prediction patch has an Intersection over Union (IoU) of at least $t_{dt}$ with the corresponding ground truth patch. In our experiments, we set this threshold to 0.4, but this value can be adjusted depending on the specific requirements of the problem. To reduce the prediction noise, we also introduced a size cutoff below which targets within the patch are not considered. This means that any patch containing only targets with a total number of pixels below the cutoff value is treated

as empty. Although this cutoff value can be arbitrarily small, we found in our experiments that a well-chosen value helps the model to focus on relevant targets, improving overall performance. The ground truth labelling process used during training is illustrated in Figure 2.

The overall model is finally trained on both tasks simultaneously. Let $x$ represent the input image, $y$ the ground truth segmentation, and $y_{cl}$ the classification ground truth based on the segmentation prediction. The global loss function is defined as:

$$
\begin{aligned}
\mathcal{L} &= \mathcal{L}_s(\hat{y}_s, y) + \alpha \mathcal{L}_{cl}(\hat{y}_{cl}, y_{cl}) \\
\hat{y}_s &= S(x, \theta_s) \\
\hat{y}_{cl} &= C(\mathbf{p}[x \oplus \hat{y}_s], \theta_c),
\end{aligned}
\tag{1}
$$

where $\mathbf{p}[\cdot]$ denotes the operation of splitting the input into patches and $\alpha$ is a tunable coefficient balancing the contributions of the two-loss components. Since the classification labels $y_{cl}$ depend on the segmentation model's predictions, the second term of the loss can directly influence the first model, leveraging the additional penalty for its errors. The two loss functions $\mathcal{L}_s$ and $\mathcal{L}_c$ can be any losses typically used for segmentation and classification tasks.

Finally, this pipeline produces the first step of segmentation prediction and the pathed correctness classification as output.

## 3.2 Dataset

We apply our framework to two datasets, encompassing 2D and 3D medical images. The first dataset (recalled as ADPKD dataset from now on), introduced by Monaco et al. in [12], is a collection of RGB immunofluorescence images of human tubules engineered from epithelial cyst-lining cells affected by ADPKD. Each image is paired with a binary image indicating the shape and size of cysts within the tissues. We adhered to the preprocessing methods described in the paper. For simplicity, we divided the dataset into a fixed train-validation-test split, ensuring that images from the same tubule were kept separate as recommended by the authors. The dataset providers committed to making it available to anyone upon request for reproducibility.

The second dataset is the Liver Tumor Segmentation Benchmark (LiTS) [2], which consists of 3D CT scans of liver tumors. For this dataset, we followed the preprocessing steps proposed in [14]. Both

datasets exhibit high variability in the size of detectable objects, and each acquisition may contain many target objects.

## 3.3 Experimental Setup

**ADPKD dataset.** For the Tandem backbones, we used a U-Net++ as the segmentation model and a ResNet18 as the classifier. The model was trained for up to 30 epochs until convergence on the validation set, using the Adam optimizer with a learning rate of $1 \times 10^{-4}$, tuned with a cosine annealing with a warm restart scheduler. The loss function $\mathcal{L}_s$ combined binary cross-entropy (BCE) and Jaccard loss, utilizing their inverse-weighted versions as proposed in [14] to make the network focus more on small-sized targets. We employed Binary Focal Loss with parameters $\alpha = 0.1$ and $\gamma = 2$ for the classification component. These parameters were determined by a grid search to correct the imbalance of labels in this subtask. The patch size used for the classification inputs is 128 pixels. This value was chosen to be large enough to capture larger cysts, but not too large to be meaningful for smaller cysts.

**LiTS Dataset.** For this dataset, the tandem backbones consisted of a 3D U-Net and a 3D ResNet18 [6]. The training pipeline for the baseline model was inspired by [14]. Specifically, we trained the models for 100 epochs using Focal Loss ($\alpha = 0.25$, $\gamma = 2$) with an inverse weight variation for $\mathcal{L}_s$, and Focal Loss ($\alpha = 0.1$, $\gamma = 2$) for $\mathcal{L}_c$. We employed the Adam optimizer with a learning rate of $1 \times 10^{-4}$, which was reduced by 20% at epochs 50 and 80. The classifier input patch size for this dataset is 32 pixels.

Following these setups, we trained the segmentation models in their baseline and tandem fashions to compare the effects of our method on segmentation performance. Then, we analyzed the classifier's performance to measure its strength in providing insights into segmentation output errors. The following section presents the results, averaging over 5 repetitions to improve statistical significance. All the experiments were conducted on a fixed train-validation-test split as provided by the datasets authors.

All experiments were implemented using the PyTorch framework on an Intel Core i9-10980XE CPU @ 3.00GHz and two NVIDIA RTX A6000 GPUs. Each experiment took approximately 2 hours for the ADPKD dataset and 12 hours for the LiTS dataset. We observed no significant variation in the training time required for baseline models and their tandem counterparts. The code used for the experiments is available online [1].

## 4 RESULTS

Table 1 reports the segmentation performance comparison of the models trained alone or within our pipeline for the two datasets. The baseline models mentioned in each dataset's respective papers are compared with their Tandem-enhanced counterparts.

Tandem models generally improve recall (Re) and intersection over union (IoU) across most size categories, with particular regard on smaller sizes, with particularly notable enhancements for small and medium-sized objects. In the ADPKD dataset, Tandem slightly improves precision (Pr) for small objects (1.85% increase), performs marginally worse for medium (3.53% decrease) and gets roughly equivalent results for large objects compared to the Base model. For recall, Tandem consistently improves performance for small

---

[1]https://github.com/simone7monaco/tantem-segmentation

(4.07% increase), medium (3.34% increase), and large objects (1.9% increase). IoU results also indicate that Tandem outperforms the Base model for small objects (5.78% increase), matches the Base model for medium objects, and slightly improves for large objects (1.03% increase).

In the LiTS dataset, Tandem shows better precision for small (14.29% increase) and medium-sized objects (14.29% increase), but a lower precision for large objects (10.05% decrease) than the Base model. Recall improvements are significant for small (19.83% increase), medium (7.91% increase), and large objects (3.58% increase) in Tandem. IoU results are better for small (15.56% increase) and medium-sized objects (14.21% increase), but slightly lower for large objects (3.38% decrease) with Tandem than the Base model. Notably, the performance on this dataset is significantly poorer than on the previous one, particularly for small and medium targets, highlighting the task's increased complexity; while this dataset is commonly used for liver detection benchmarks where models perform well, few studies focus on tumor detection.

Overall, the Tandem model effectively enhances recall and IoU, which are crucial for medical segmentation tasks, particularly for smaller and medium-sized objects. However, there is a trade-off with a slight reduction in precision for larger objects, suggesting a need for further tuning or model adjustments to achieve optimal performance across all size categories. Finally, considering segmentation capability alone, it's important to note that some of the observed variations are relatively minor, indicating that the performance of a network trained with the Tandem procedure is quite comparable to one trained with a baseline approach. The crucial distinction, however, is that the Tandem method not only avoids performance degradation but also offers slight improvements, while additionally providing a self-evaluation of the model predictions.

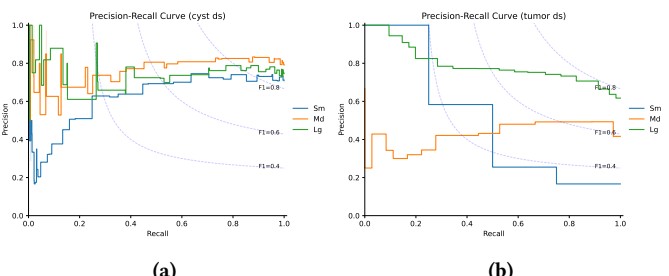

**(a)**          **(b)**

**Figure 3: P-R curves for the Tandem confidence maps both on LiTS and ADPKD datasets.**

To quantify the Tandem method's capacity for providing reliable confidence maps, Figure 3 displays the precision-recall curve of the Tandem models on the classification subtask, categorized by target sizes. This plot illustrates the tradeoff between the precision and recall metrics on test patches as the classifier threshold varies to determine whether a prediction is considered *correct* or *wrong*. The dashed lines represent the ISO-F1 curves, indicating points on the plane associated with specific F1 scores. Each patch is labelled according to the size category of the object it contains, as per the ground truth.

**Table 1: Segmentation results for dataset and target size**

| | | Pr | | | Re | | | IoU | | |
|---|---|---|---|---|---|---|---|---|---|---|
| | | Small | Medium | Large | Small | Medium | Large | Small | Medium | Large |
| cyst | Base | 0.485±0.056 | **0.764±0.046** | **0.799±0.023** | 0.664±0.109 | 0.748±0.049 | 0.685±0.017 | 0.381±0.022 | **0.605±0.013** | 0.584±0.011 |
| | Tandem | **0.494±0.022** | 0.737±0.032 | 0.795±0.044 | **0.691±0.058** | **0.773±0.030** | **0.698±0.039** | **0.403±0.010** | 0.605±0.005 | 0.59±0.022 |
| LiTS | Base | 0.049±0.011 | 0.224±0.016 | **0.537±0.042** | 0.363±0.052 | 0.556±0.030 | 0.781±0.035 | 0.045±0.010 | 0.190±0.010 | **0.467±0.035** |
| | Tandem | **0.056±0.015** | **0.256±0.039** | 0.483±0.059 | **0.435±0.082** | **0.600±0.046** | **0.809±0.019** | **0.052±0.014** | **0.217±0.026** | 0.433±0.045 |

In the ADPKD dataset, the model performs best for medium-sized targets, followed closely by large and small sizes. The performance across all three sizes is comparable, indicating that the confidence maps generated by the Tandem method are stable and reliable regardless of the detected cyst size. The three curves exhibit a slightly horizontal orientation, with the medium-sized targets achieving a maximum precision of 0.8 as they approach the best recall values. Large and small targets follow closely, reaching around 0.7 maximum precision. This horizontal orientation of the PR curves suggests that the classifier maintains relatively high precision across a wide range of recall values, indicating a balanced performance in detecting positive and negative instances. The high F1 values achieved in these regions of the plot further underscore the overall effectiveness of the classifier.

In the tumor dataset, the model shows the highest performance for large-sized targets, with the corresponding PR curve positioned at the top. This curve is not completely flat but slopes downward slightly. When the F1 score reaches 0.8, the recall is approximately 0.9, and the precision is around 0.8. This indicates robust performance for large tumors, maintaining a high balance between precision and recall, and effectively managing both true positive detections and minimizing false positives.

The PR curve for medium-sized targets is slightly lower and exhibits a moderate increase. Although it does not achieve an F1 score of 0.8, it reaches its best performance with an F1 score above 0.6, with both precision and recall around 0.6 at the highest recall values. This indicates a reasonable performance for medium-sized tumors, but highlights that there is still room for improvement to achieve higher precision and recall.

For small-sized targets, the PR curve shows a steep decline, with the best performance, or "knee," occurring at an F1 score of about 0.5, with recall at 0.5 and precision at 0.6. This suggests that detecting small tumors remains a challenging task for the model, as indicated by the lower F1 scores and the sharper drop in performance metrics.

Overall, the variations in performance across different target sizes in the tumor dataset highlight the increased complexity of this detection task. Despite these challenges, the high F1 scores achieved in specific regions of the plot further reinforce the classifier's overall effectiveness.

## 5 CONCLUSIONS

The results presented in Section 4 and Table 1 clearly demonstrate the effectiveness of the Tandem model in improving the precision, recall, and IoU across different object sizes in the cyst and LiTS datasets. The Tandem model shows improvements, particularly in managing smaller and medium-sized targets, which are crucial for medical segmentation tasks.

In terms of segmentation capability, while some variations are relatively minor, the improvements in critical metrics such as recall and IoU emphasize the practical benefits of our approach. Moreover, the ability of the Tandem model to provide reliable confidence maps as shown in Figure 3, particularly in differentiating between tumor sizes in the LiTS dataset, adds a layer of validation to the model's predictive accuracy and reliability. Ultimately, this study showcases the potential of the proposed model in enhancing medical image analysis, providing valuable insights to physicians.

## ACKNOWLEDGMENTS

Acknowledgments are very welcome here

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
