# OpenReview forum: "Tandem: a Confidence-based Approach for Precise Medical Image Segmentation"
_KDD.org/2024/Workshop/AIDSH — KDD-AIDSH 2024 Poster_

### Official Review · Reviewer_ZMiu · 2024-06-18
**This work proposed Tandem, a segmentation model for kidney cysts and tumor masses. However, the innovation and experimental results are not sufficient.**

**Rating:** 4
**Confidence:** 4

**Review:**

Pros:
This work proposed a new architecture of medical imaging segmentation, especially for kidney cysts and tumor masses. Experimental results on two datasets, to some extent, demonstrated the effectiveness of this approach.

Cons:
1. The overall architecture is not novel, the classifier is based on the input of the image and its corresponding prediction, which is more like a simple attempt.
2. The results are not sufficient. As shown in Tab.1, no other state-of-the-arts are included. Besides, the improvement over the base model is marginal.
3. The third contribution point declared a practical evaluation of the proposed model, which I did not find in the paper.

---

### Official Review · Reviewer_tBWG · 2024-06-21
**Submission 27**

**Rating:** 6
**Confidence:** 5

**Review:**

This article proposed a method named “Tandem Analysis for Neural Detection and Evaluation Model” for kidney diseases segmentation and classification. In detail, it contains a segmentation network and a classification network. The classification network takes the original image and the segmentation result as the input, and gives out the confidence score of the segmentation results. However, there exists many issues:
Major concerns:
1. In the introduction, the primary goal of the article is ambiguous. As this paper want to solve the problems of “small targets identification”, “providing a mechanism to measure the model’s confidence”, “investigating the effect of this innovative refinement strategy applied to state-of-the-art deep-learning solutions to improve the segmentation of variablesized and scattered medical targets.”. The logic of the motivation is not organized well.
2. The Autosomal Dominant Polycystic Kidney Disease (ADPKD) segmentation is a well studied field and there exists many relevant works[1-3], this article did not investigate these papers and lacks proper citations and introduction when reviewing the related works.
3. The unfold operation is not explained in Figure 1. And the output of the classifier is not presented. This makes the reader confusing.
4. The workflow of the method is not clear, an overview figure is needed.
5. In experimental settings, the train-val-test split is not given. And the train-inference workflow is not well described.
6. How are the targets divided into small, medium and large?
7. The experiment lacks the ablation study. What’s the base model? How is it chosen?
8. Table 1 lacks the discussion that the based model outperforms the proposed method.
9. This article has no visualization of the segmentation results.

Minor:
1. The data format should be unified, with two decimal places retained in Table 1.

[1]	Woznicki, Piotr, et al. "Automated kidney and liver segmentation in MR images in patients with autosomal dominant polycystic kidney disease: a multicenter study." Kidney360 3.12 (2022): 2048-2058.
[2]	Goel, Akshay, et al. "Deployed deep learning kidney segmentation for polycystic kidney disease MRI." Radiology: Artificial Intelligence 4.2 (2022): e210205.
[3]	Monaco, Simone, et al. "AI models for automated segmentation of engineered polycystic kidney tubules." Scientific Reports 14.1 (2024): 2847.

---

### Decision · Program_Chairs · 2024-06-28

Accept (Poster)